# The Role of Iron Metabolism in Fatigue, Depression, and Quality of Life in Multiple Sclerosis Patients

**DOI:** 10.3390/ijerph17186818

**Published:** 2020-09-18

**Authors:** Anna Knyszyńska, Aleksandra Radecka, Paulina Zabielska, Joanna Łuczak, Beata Karakiewicz, Anna Lubkowska

**Affiliations:** 1Department of Functional Diagnostics and Physical Medicine, Pomeranian Medical University in Szczecin, 71-210 Szczecin, Poland; aleksandra.radecka@pum.edu.pl (A.R.); anna.lubkowska@pum.edu.pl (A.L.); 2Subdepartment of Social Medicine and Public Health, Department of Social Medicine, Pomeranian Medical University in Szczecin, 71-210 Szczecin, Poland; paulina.zabielska@pum.edu.pl (P.Z.); beata.karakiewicz@pum.edu.pl (B.K.); 3Faculty of Health Sciences, College of Engineering and Health in Warsaw, 02-366 Warsaw, Poland; joasialuczak@op.pl; 4Department of Cardiological Rehabilitation, Central Clinical Hospital of the Ministry of Internal Affairs and Administration in Warsaw, 02-507 Warsaw, Poland

**Keywords:** chronic fatigue syndrome, depression, iron, multiple sclerosis, quality of life

## Abstract

Multiple sclerosis (MS) is a chronic inflammatory disease of autoimmune origin for which there is currently no available cure. In the course of MS, next to neurological disorders, patients often present with chronic fatigue syndrome and depressive disorders, which impact on their daily function and quality of life. The aim of study was to analyse the relationship between serum parameters of iron metabolism and the severity of fatigue, depressive symptoms, and quality of life in MS patients. Methods: The study sample consisted of 90 people with a diagnosis of multiple sclerosis, age range 19–67 years, whose functional status evaluated using the Expanded Disability Status Scale in 90% of the participants did not exceed 3.5 points. Venous blood samples were collected for blood cell count determination and for the purposes of obtaining serum analysed for the concentrations of iron, ferritin, transferrin, transferrin saturation, unsaturated iron binding capacity (UIBC), and total iron binding capacity (TIBC). The participants were also evaluated according to the Fatigue Severity Scale, Beck Depression Inventory, and Functional Assessment of Multiple Sclerosis. Results: Ferritin levels were significantly correlated with the severity of depressive symptoms (r = −0.22; *p* = 0.04) and quality of life assessment (r = 0.22; *p* = 0.04) in the MS patients. Moreover, the severity of fatigue and depressive symptoms was significantly linked to a deterioration in quality of life. Conclusions: Ferritin deficiency in MS patients is associated with an exacerbation of depressive disorders and a decline in quality of life. Symptoms of fatigue in MS patients are inversely proportional to mood and quality of life.

## 1. Introduction

Multiple sclerosis (sclerosis multiplex, MS) is a chronic inflammatory disease of autoimmune origin for which there is currently no available cure. It affects the central nervous system, with multiple focal demyelinating lesions forming over time in the brain and the spinal cord [1]. Fatigue is regarded as one of the most common symptoms in the course of MS. According to different studies, it affects 53–95% of MS patients. In more than half, it is the predominant symptom, while 70–95% identify chronic fatigue as one of the three main complaints [2]. What is more, chronic fatigue is often reported as the first symptom observed by MS patients, even before their condition is diagnosed [3]. Depressive symptoms and major depression are also some of the most common comorbidities of multiple sclerosis (MS), and at the same time they are an integral part of neurological symptoms and functional impairment, measured using the Expanded Disability Status Scale (EDSS) [4]. The high prevalence and severity of depressive symptoms in MS sufferers has a powerful effect on their function in everyday life, as one of the main factors determining patients’ quality of life [5]. Iron in the human body is found in haemoglobin, myoglobin, digestive enzymes, and in storage form (ferritin). Its role is related particularly with tissue respiration processes. Iron participates in a wide range of biochemical processes necessary for the normal function of the brain, including as a cofactor for enzymes involved in the metabolism of neurotransmitters and myelin [6]; as a component of cytochromes it is part of the electron transport chain [7]. Iron deficiency is associated with a weakness and fatigue and decreased physical and cognitive performance, with worse outcomes in patients with coexisting pathologic conditions [8,9]. Conversely, excess iron (iron overload) has toxic effects, because it catalyses the formation of reactive oxygen species, resulting in oxidative stress, which has been implicated in the pathogenesis of neurodegenerative diseases including multiple sclerosis [10,11]. Many chronic neurological disorders, including MS, have been shown to be accompanied by disorders of iron metabolism, often related to its increased deposition [12,13,14,15]. Considering the role of iron in the body, it is no wonder that scholars seek answers as to whether disorders of iron metabolism may be one of the factors triggering and modulating disease development, or whether they are in fact secondary to the disease process. To date, this question has not been unequivocally settled. The evidence of iron deposition in the body in the course of MS comes primarily from patient brain-imaging studies employing magnetic resonance imaging [16,17]. Despite these observations, the space–time dynamics of iron deposition and the cellular pathways involved in internal iron accumulation in MS have not yet been elucidated. A major limitation in our understanding of the role of iron in MS is that most of the knowledge comes from the histopathology of long-term patients [18]. Over the past few years, several scholars have set out to determine the levels of iron, ferritin, and transferrin, both in cerebrospinal fluid and blood of MS patients. However, the findings published to date are not consistent, sometimes to the point of contradicting each other. There are several reports in the literature analysing the relationship between the parameters of iron metabolism and the intensity of fatigue or the occurrence of depressive disorders in post-stroke patients [19], Parkinson’s disease [20], in pregnant women [21], in diabetes [22], and in fibromyalgia syndrome [23], but there are no such studies among patients with MS.

The aim of this study was to analyse the relationship between serum parameters of iron metabolism and the severity of fatigue, depressive symptoms, and quality of life in multiple sclerosis patients.

## 2. Materials and Methods

### 2.1. The Study Group

The material used in this study was collected from April to July 2018 at John Paul II Multiple Sclerosis Rehabilitation Centre in Borne Sulinowo, MSWiA Central Teaching Hospital in Warsaw, and the Tomasz Sokołowski Teaching Hospital of the Pomeranian Medical University in Szczecin.

Each of the volunteers qualified to participate in the study was informed in detail about:Research schedule and research procedures,Voluntary participation in the study and the possibility of resigning from further participation at any time of its duration, or not consenting to the use of their results in scientific works,Anonymity of results that will be used only for scientific purposes.

All participants signed a written informed consent form, and the study was conducted in accordance with the Helsinki Declaration. The protocol was approved by the Bioethics Committee of the Pomeranian Medical University (KB-0012/34/15).

Inclusion criteria: Age ≥18 years, diagnosed by a neurologist with multiple sclerosis, functional status assessed on the EDSS scale by a neurologist at <6 points, and no iron supplementation. People who declared iron supplementation and whose functional state did not allow for independent walking (>6 points on the EDSS scale) were not included in the study. Women who were in the perimenstrual period on the day of the study were also not included in the research.

The study group consisted of 90 people (62 women and 28 men) with a diagnosis of multiple sclerosis, aged 19–67 years. The average duration of disease in the study group was 8.22 (6.21), (median, Me = 6) years. On average, the women had had the condition for one year longer than the men. Among both women and men, the time since the diagnosis fell in the range of 3–12 years (Me = 7 years for women and 6 years for men). A substantial majority of participants (78%) had the relapsing-remitting form of multiple sclerosis (RRMS) (48 women and 22 men). The secondary-progressive type (SPMS) was found in 14 people (8 women and 6 men), and the primary-progressive type (PPMS) affected only 6 women and not a single man. On the Expanded Disability Status Scale (EDSS), the participants’ median score was 3 points. More than 90% of the participants (57 women and 26 men) scored no more than 3.5 points.

### 2.2. Biochemical Determination of Blood

In the course of the study, on an empty stomach after a 15-min rest, a single venous blood sample was collected from the antecubital vein in each person, for the purposes of blood cell count determination (8% K3 EDTA solution) and obtaining serum (tubes coated with coagulation activator) using the VACUETTE evacuated blood collection system. The following parameters were determined in blood serum: Iron concentration [µg/dL], UIBC (unsaturated iron binding capacity) [µg/dL], TIBC (total iron binding capacity) [µg/dL], transferrin saturation [%], transferrin [mg/dL], and ferritin [μg/L].

### 2.3. Questionnaire Surveys

Data on chronic fatigue syndrome, depressive symptoms, and quality of life were collected using the diagnostic survey method, with appropriate standardized questionnaire surveys.

The severity of fatigue was evaluated using the Fatigue Severity Scale (FSS), which is currently one of the most frequently used inventories for measuring fatigue. The scale contains nine simple statements concerning one’s functioning over the past week, and the patients are asked to rate each one on a scale from 1—“strongly disagree” to 7—“strongly agree”. The statements included in the questionnaire cover the effects of fatigue on various aspects of daily living, related to both mental and physical functioning. The final score is the arithmetic mean of the points scored in individual items. The following classification was adopted in the interpretation of the scores: No fatigue (FSS ≤ 4.0 pts), borderline fatigue (4.0 pts < FSS < 5.0 pts), and fatigue (FSS ≥ 5.0 pts) [24,25]. Cronbach’s alpha for the Polish version of FSS = 0.89 [26].

The intensity of depression was evaluated using the Beck Depression Inventory (BDI-II), which is designed to measure the severity of depressive symptoms, especially to detect depression in screening studies of larger populations, but it may also be used for assessing the progression of depressive disorders and measuring response in the course of treatment. The majority of symptoms included in BDI-II reflect the diagnostic criteria for depression according to the DSM-IV. The inventory contains 21 items representative of individual symptoms of depression, the severity of which is rated on a 4-point scale. The total score is the sum of points scored for all 21 symptoms. The highest possible score is 63 points. A score of 0–9 points is categorised as no depression, 10–19 points is mild depression, 20–25 points is moderate depression, and 26–63 points is severe depression [27]. Cronbach’s alpha for the Polish version of BDI_II = 0.93 for healthy people, and 0.95 for people with depressive disorders [28].

Quality of life was evaluated with reference to the Functional Assessment of Multiple Sclerosis (FAMS). The instrument consists of 59 questions, divided into the principal test (44 items) and miscellaneous items evaluating “additional concerns” (15 items). Questions from the principal test, based on factor analysis, have been divided into six subscales, evaluating: Mobility (7 questions), symptoms related to MS progression and treatment (7 questions), emotional status related to the patient’s mood and feelings (7 questions), contentment in everyday life (7 questions), quality of cognitive function, referred to generally as thinking and fatigue (9 questions), and family and social well-being (7 questions). The other 15 questions referring to “additional concerns” evaluate items like sexual well-being and side effects of MS treatment, and are not included in the overall score [29]. The total score range is between 0 and 176 points, with higher scores indicative of better quality of life assessment. Brola et al. (2007) proposed the following interpretation of FAMS scores, distinguishing three levels of quality of life: 0–57 points—poor; 58–117 points—moderate, 118–176 points—good quality of life [30]. The Polish version of the FAMS questionnaire was subject to standardization and was found to be a valuable psychometric tool with satisfactory external validity (Cronbach’s alpha = 0,95) [29].

All the surveys were self-administered by the respondents, completed at their own pace, on the on the same day when the blood was collected. If the respondent had any problems understanding a given question or statement included in any survey instruments, they had an opportunity to ask for clarification.

### 2.4. Statistical Analysis

Statistical analysis was performed with STATISTICA computer software (version 13.3, TIBCO Software, Palo Alto, CA, USA). Elements of descriptive statistics and statistical inference were employed. Data distribution was tested for normality using the Shapiro–Wilk test. The significance of differences between groups in terms of selected variables was evaluated using tests for independent samples. With normally distributed data, Student’s t-test was used. In turn, differences between data following a non-normal distribution were tested for significance using the non-parametric Mann–Whitney U test. Correlations between variables with a non-normal distribution were identified using a non-parametric test, namely Spearman’s rank correlation coefficient (rho). The statistical significance level was set at *p* < 0.05.

## 3. Results

Following the analysis of blood biochemistry determinations, it was concluded that the medians and means of the tested blood parameters fell within the reference range for the general adult population [31]. With regard to results falling outside of the normal range, the highest proportion of the study group, namely 32 people, presented abnormal serum transferrin saturation. This was followed by iron determinations [μg/dL], with 19 people having abnormally low or high results. The determinations of unsaturated (UIBC) and total (TIBC) iron binding capacity deviated from the normal range in 16 and 11 patients, respectively. In the following parameters, the adopted reference ranges are the same for men and women: Iron, UIBC, TIBC, transferrin saturation, and transferrin. In all of these variables, with the exception of TIBC, significant differences were observed between men and women. Women’s results were significantly higher than men’s for UIBC (*p* = 0.02) and transferrin (*p* = 0.04) (Table 1).

A general analysis of the median scores in the FSS questionnaire, with reference to the adopted cut off points, suggests that the severity of fatigue symptoms among the participants does not warrant a diagnosis of chronic fatigue syndrome. Both men and women had scores in a similar range (1–7 points), but the median in both groups was <4 points. Looking at the results in more detail, out of all the participants, 24 people (18 women and 6 men) showed symptoms of fatigue, with FSS scores >5 points. Moreover, 16 people (10 women and 6 men) had scores indicative of borderline fatigue. However, more than half of the participants (34 women and 16 men) were not found to display particularly severe symptoms of fatigue, which could justify a diagnosis of chronic fatigue syndrome (FSS ≤ 4 points) (Table 1)**.**

The range of scores obtained in the BDI-II survey points to a varying intensity of depressive symptoms in the study group. The mean scores for the study group as a whole and for the men’s group were in the range indicative of the absence of depression, 9.42 (9.00) points for all participants, and 7.64 (8.44) points for men. The mean BDI-II scores in the women’s group may be interpreted as indicative of mild depression,10.23 (9.13) points. However, with such a high dispersion of scores, it is advisable to consider the median scores, which both for the group as a whole, and in gender-specific groups, were in the points range suggesting the absence of depression. The differences in the number of points scored by men and women were not statistically significant (*p* = 0.07) (Table 1).

None of the participants reported poor quality of life, with no scores ≤57 points in the entire study group. Among men, only nine had scores falling into the moderate range, while the others reported good quality of life. The numbers of women reporting moderate and good quality of life was comparable, with a few more results in the lower range. The median score in the FAMS questionnaire seems to suggest that the quality of life in the study group was good (118 points). Statistically significant differences were observed between the FAMS scores for men and women (*p* = 0.04), suggesting that the quality of life assessment among women was moderate (111 points), and good in men (124 points). The participants’ scores in the respective FAMS subscales were similar and for the most part did not point to gender-specific differences. The scores in the “thinking and fatigue” domain were an exception here, with women’s scores (Me = 19 points) significantly lower than men’s (Me = 24.5 points). Likewise, differences were also observed in the analysis of “additional concerns”, but these are not included in the overall score (Table 1).

The analysis of correlations between blood biochemical parameters and those related to the degree of disability, fatigue severity, symptoms of depression, and quality of life assessment revealed few significant relationships. Only ferritin levels were significantly correlated with the severity of depressive symptoms (r = −0.22; *p* = 0.04) and quality of life assessment (r = 0.22; *p* = 0.04) in the patients. A significant positive correlation was found for the intensity of depressive symptoms vs. the degree of disability and severity of symptoms of chronic fatigue syndrome. This suggests that in the study group, the exacerbation of fatigue symptoms and increasing disability was accompanied by a statistically significant increase in depressive symptoms (*p* < 0.001). Moreover, the findings show a significant negative correlation between BDI-II scores and FAMS results (r = −0.56, *p* < 0.001). It may therefore be concluded that the intensity of depressive symptoms was associated with quality of life impairment across all the domains included in the analysis. It was also observed that impaired mobility and increasing fatigue among the participants co-occurred with lower quality of life scores (*p* < 0.001). At the same time, the analysis of the results demonstrated that the level of disability did not have a significant effect on the prevalence of the chronic fatigue syndrome among the participants (*p* = 0.06) (Table 2). 

## 4. Discussion

The present study focused on the assessment of iron metabolism in 90 MS patients. The patients’ blood test results were compared to reference ranges adopted for the general population, taking into account age and gender. Despite sizeable individual differences, notably in transferrin saturation and iron levels, it may be concluded that the mean values for all the iron metabolism parameters included in the study fell within the normal range. Findings from this study did not confirm the relationship between iron metabolism parameters in serum and the duration of disease or degree of disability according to EDSS, however the participants’ age was found to be significantly positively correlated with ferritin (r = 0.25; *p* = 0.02) and significantly negatively correlated with TIBC (r = −0.23; *p* = 0.03). In this study, increased ferritin concentrations were found in SPMS patients (Me = 123 µg/L) compared to RRMS patients (Me = 59 µg/L). However, due to the disproportionate numbers of patients representing the respective types of MS, it was difficult to carry out a robust analysis of the relationship between blood biochemical parameters and disease course. Consequently, this result should be regarded exclusively as an indication of a trend that should be verified in further research. In particular that in the literature, there are some reports of associations between the course of MS and ferritin content in the patients’ blood and/or cerebrospinal fluid. The values are significantly higher in patients suffering from chronic progressive MS compared to the RRMS group [32,33,34]. Findings from this study may be regarded as a confirmation of the results from earlier research by Reider et al. [14] and Iranmanesh et al. [15], where no differences were observed in iron and ferritin levels between groups of MS patients and healthy individuals. Likewise, Visconti et al. [35], in their study with the participation of 12 patients after the first episode of MS, found no significant differences in serum iron levels of the patients, both in the acute phase of the episode and at a six-month follow-up, compared to healthy controls. Findings from studies carried out in Greece and Egypt [32,36] also failed to show any differences in iron content, but upon detailed analysis of test results elevated transferrin levels were observed in MS patients compared to people without MS. At this point, it is worth bringing up the study by Iranmanesh et al. [15], where no relationship was revealed between serum ferritin levels and age, gender, or disease course. Similar correlations were observed by other authors [32,36]. However, several authors showed increased or decreased serum iron and/or ferritin levels in those with the disease [16,37]. Forte et al. [37] in their study of 60 MS patients demonstrated lower serum iron concentrations in the patients compared to a group of healthy individuals, while Johnson et al. observed an inverse relationship [16]. Le Vine et al. noted higher ferritin levels in the cerebrospinal fluid of MS patients compared to individuals not affected by MS [33]. Ferreira et al. [38] evaluated serum ferritin levels and confirmed their correlation with oxidative stress markers and MS progression, suggesting that elevated concentrations of ferritin may increase oxidative stress in multiple sclerosis patients and contribute to disease progression The contradictions in the obtained results may be caused by different criteria for selecting the test and control groups. Perhaps the decreased blood levels of iron and ferritin in MS patients are associated with longer disease duration, when iron begins to accumulate more in the brain [39]. This hypothesis should be tested by conducting research among larger groups of patients with SI similar to each other with the duration of the disease.

It has been put forward that the key factors contributing to neurodegeneration in MS patients, apart from the activation of microglia, chronic oxidative damage, and mitochondrial alterations in axons, may also include iron accumulation in the human brain with age [40]. Iron release from oligodendrocytes and myelin due to the upregulated activity of ferroxidases leads to its extracellular accumulation in patients’ brain tissue and uptake by microglia and macrophages, resulting in degenerative lesions [41]. Undoubtedly, changes in the brain can influence mood disorders. To date, there have not been many studies assessing the relationships between iron metabolism parameters and chronic fatigue syndrome, depressive symptoms, and functional status in MS patients and also their quality of life. This study is an attempt at such an analysis, which demonstrated that blood biochemical parameters are only poorly correlated with the variables included in the analysis. In this study, patients with more severe depressive symptoms had lower ferritin levels (r = −0.22; *p* = 0.04) and were also found to have lower haemoglobin. The present findings lend themselves to the conclusion that elevated serum ferritin levels in MS patients have a positive effect on mood, showing a positive correlation with lower rates of depression (BDI-II, r = −0.22; *p* = 0.04), and at the same time better quality of life (FAMS, r = 0.22; *p* = 0.04), especially in the domain of mental health. There is, however, evidence in the literature that depression is accompanied by alterations in iron metabolism, including decreased levels of iron and transferrin, as well as elevated levels of serum ferritin, reduced number of erythrocytes, and lower haematocrit and haemoglobin [42,43,44]. Zhu et al. [19] published a report that demonstrated elevated ferritin levels in people with post-stroke depression. In turn, relationships similar to those obtained by us were presented in studies conducted among healthy 312 men and 216 women working in municipal offices in Japan. In men, decreased levels of serum ferritin were significantly correlated with a higher prevalence of depressive symptoms [45]. Another study carried out among a young population also demonstrated significantly lower ferritin concentrations corresponding with increased prevalence of depression [46]. However, these studies were conducted on people who did not suffer from MS. Kallaur et al. [4] analysed, among other things, the link between depression and blood ferritin concentrations in MS patients, as one of the few so far. Their findings showed a significantly higher ferritin level in MS patients compared to healthy controls, but in the analysis of results of patients with and without depressive symptoms, no differences were found between the groups in terms of ferritin concentrations. Seeing as how there are no more reports unequivocally confirming such correlations with regard to MS patients, further research is needed.

Another objective of this study was to pinpoint the link between the levels of iron and iron-regulating parameters in the blood of MS patients vs. chronic fatigue, a symptom that appears to be an integral part of the disease. Due to the functions of iron in the body, it was assumed that its deficiency will increase fatigue severity. However, our findings did not confirm this hypothesis. No relationship was identified between the values of blood cell count and biochemical parameters vs. the severity of fatigue symptoms. Please note that the level of fatigue reported by the patients included in the study group was low, and their age range was wide, which may have had a modulating effect on the results. To our knowledge, in the available literature, there are no studies analysing correlations of this nature, suggesting a need for continued research with more numerous study samples, which would allow for a verification of the hypothesis underlying this study adjusted for age, duration of disease, degree of disability, and fatigue. There are valid reasons to undertake research in larger patient groups, taking significant symptoms of chronic fatigue syndrome as an inclusion criterion.

### 4.1. Limitations of the Study

The biggest limitation of our research is the lack of a control group of healthy people, but on the other hand, the study was aimed at presenting the relationship between blood biochemical results and the level of fatigue and depressive disorders in a group of patients with MS.

Our studies mostly involved people without major symptoms of fatigue and depression, so we would like to continue our study, taking into account the severe chronic fatigue syndrome and depression as the inclusion criteria.

### 4.2. Practical Implications

Knowledge about the relationship between the values of the parameters of iron metabolism in the blood of MS patients and their mood and the intensity of fatigue may affect the possibility of their regulation and thus improve the quality of life of people with multiple sclerosis.

## 5. Conclusions

Ferritin deficiency in multiple sclerosis patients is associated with an exacerbation of depressive disorders and a decline in quality of life. Symptoms of fatigue in MS patients are inversely proportional to mood and quality of life.

## Figures and Tables

**Table 1 ijerph-17-06818-t001:** Mean ± SD, median, min–max, lower and upper quartile in blood biochemical parameters, survey results, and significance of gender-specific differences.

Variables			*n*	Mean ± SD	Me	Min–Max	Q1–Q3	*p*
**AGE**		All	90	42.5 ± 11.96	48.5	19.0–67.0	32.3–50.0	-
		Female	62	43.4 ± 12.09	42.5	19.0–67.0	34.3–51.0	0.41
		Male	28	41.2 ± 11.98	38.5	19.0–64.0	32.0–48.5	
**EDSS**		All	90	2.9 ± 0.90	3.0	1.5–6.5	2.5–3.0	-
		Female	62	3.0 ± 0.90	3.0	1.5–5.5	2.5–3.0	0.45
		Male	28	2.8 ± 1.00	2.3	1.5–6.5	2.0–3.0	
**Iron Metabolism**	**Iron, μg/dL**	All	90	93.3 ± 40.55	88.8	11.4–272.0	67.2–111.7	-
		Female	62	86.9 ± 42.33	80.7	11.4–272.0	59.9–103.6	0.04
		Male	28	107.7 ± 30.99	102.1	63.6–197.3	84.3–118.3	
	**UIBC, μg/dL**	All	90	220.7 ± 82.25	220.5	12.0–438.4	160.1–268.2	-
		Female	62	234.9 ± 88.07	229.1	12.0–438.4	170.4–287.5	0.01
		Male	28	189.3 ± 54.76	193.4	60.5–275.0	154.5–231.5	
	**TIBC, μg/dL**	All	90	314.9 ± 62.86	316.6	166.3–449.8	271.9–350.2	-
		Female	62	322.0 ± 65.20	324.4	198.3–449.8	272.5–374.0	0.16
		Male	28	299.0 ± 52.73	303.7	166.3–402.0	271.8–336.2	
	**Transferrin saturation, %**	All	90	31.6 ± 15.41	30.0	2.5–95.6	22.7–39.6	-
		Female	62	29.1 ± 16.26	27.1	2.5–95.6	18.6–37.5	0.00
		Male	28	37.3 ± 11.07	37.1	18.8–65.4	30.9–41.6	
	**Transferrin, mg/dL**	All	90	268.5 ± 43.63	261.0	181.4–368.0	236.8–299.3	-
		Female	62	275.5 ± 45.26	273.5	194.0–368.0	237.8–308.5	0.04
		Male	28	252.9 ± 34.13	252.5	181.4–353.0	229.5–272.0	
	**Ferritin, μg/L**	All	90	101.5 ± 106.9	64.1	3.3–570.0	29.3–126.3	-
		Female	62	59.5 ± 55.19	49.7	3.3–349.0	21.0–82.3	0.00
		Male	28	194.7 ± 130.34	199.8	50.6–570.0	93.8–235.0	
**Fatigue** **FSS**		All	90	4.0 ± 1.50	3.9	1.1–7.0	3.0–5.0	-
		Female	62	4.0 ± 1.53	3.8	1.1–7.0	2.9–7.2	0.75
		Male	28	4.1 ± 1.42	3.9	1.7–6.9	3.1–4.7	
**Depression** **BDI-II**		All	90	9.4 ± 9.0	6.5	0–37.0	2.5–15.0	-
		Female	62	10.2 ± 9.13	7.0	0–37.0	3.0–15.0	0.07
		Male	28	7.6 ± 8.44	4.0	0–31.0	1.0–15.0	
**Quality of Life**	**FAMS Overal**	All	90	116.6 ± 24.08	118.0	65.0–169.0	98.3–133.8	-
		Female	62	112.4 ± 41.80	111.0	65.0–154.0	96.5–128.8	0.04
		Male	28	124.1 ± 27.08	126.0	67.0–169.0	101.8–148.5	
	**FAMS–M**	All	90	18.0 ± 6.03	19.0	3.0–28.0	14.0–23.0	-
		Female	62	17.6 ± 5.88	17.5	4.0–28.0	14.0–23.0	0.33
		Male	28	18.9 ± 6.27	20.0	3.0–28.0	14.0–23.3	
	**FAMS–S**	All	90	18.9 ± 5.49	18.0	8.0–28.0	15.0–24.0	-
		Female	62	18.2 ± 5.54	17.0	8.0–28.0	14.0–23.0	0.06
		Male	28	20.6 ± 4.98	21.0	12.0–27.0	15.8–25.3	
	**FAMS–EWB**	All	90	19.6 ± 6.31	20.0	4.0–28.0	15.0–24.0	-
		Female	62	19.2 ± 6.04	20.0	4.0–28.0	15.0–23.0	0.32
		Male	28	20.5 ± 6.77	23.0	6.0–28.0	14.8–27.0	
	**FAMS–GC**	All	90	19.6 ± 5.56	20.0	5.0–28.0	15.3–23.8	-
		Female	62	19.5 ± 4.97	20.0	6.0–28.0	16.3–23.0	0.57
		Male	28	19.9 ± 6.68	21.0	5.0–28.0	14.0–27.0	
	**FAMS–TF**	All	90	20.4 ± 7.91	20.0	4.0–35.0	15.0–26.7	-
		Female	62	19.0 ± 7.98	19.0	4.0–35.0	14.3–25.0	0.01
		Male	28	23.7 ± 6.69	24.5	7.0–35.0	19.0–29.0	
	**FAMS–FSWB**	All	90	19.5 ± 5.68	20.0	5.0–28.0	15.3–24.0	-
		Female	62	19.1 ± 5.99	20.0	5.0–28.0	15.0–23.0	0.36
		Male	28	20.6 ± 4.77	20.0	12.0–28.0	17.0–24.3	
	**FAMS–AC**	All	90	35.8 ± 8.25	37.0	14.0–52.0	29.3–41.0	-
		Female	62	34.5 ± 8.44	33.5	14.0–52.0	28.0–40.0	0.03
		Male	28	38.6 ± 7.02	39.0	23.0–51.0	33.0–43.3	

Abbreviations: EDSS, Expanded Disability Status Scale; FSS, Fatigue Severity Scale; BDI-II, Beck Depression Inventory; FAMS, Functional Assessment of Multiple Sclerosis; M, mobility; S, symptoms; EWB, emotional well-being; GC, general contentment; TF, thinking and fatigue; FSWB, family and social well-being; AC, additional concerns.

**Table 2 ijerph-17-06818-t002:** Spearman rank correlations between the analysed variables.

		EDSS		Quality of Life FAMS		Fatigue FSS		Depression BDI-II	
		rho	*p*	rho	*p*	rho	*p*	rho	*p*
**Iron Metabolism**	**Iron, μg/dL**	−0.02	0.87	0.16	0.12	0.01	0.90	−0.12	0.25
	**UIBC, μg/dL**	−0.04	0.69	−0.09	0.41	0.11	0.32	0.15	0.17
	**TIBC, μg/dL**	−0.15	0.17	−0.01	0.96	0.14	0.19	0.06	0.55
	**Transferrin saturation, %**	−0.00	0.99	0.14	0.18	−0.07	0.49	−0.18	0.09
	**Transferrin, mg/dL**	−0.04	0.69	−0.07	0.49	0.18	0.09	0.03	0.79
	**Ferritin, μg/L**	0.03	0.80	0.22	0.04	−0.18	0.09	−0.22	0.04
**Depression—BDI-II**		0.28	0.01	−0.56	0.00	0.51	0.00	-	-
**Fatigue—FSS**		0.20	0.06	−0.47	0.00	-	-	-	-
**Quality of life—FAMS**		−0.50	0.00	-	-	-	-	-	-

Abbreviations: EDSS, Expanded Disability Status Scale; FSS, Fatigue Severity Scale; BDI-II, Beck Depression Inventory; FAMS, Functional Assessment of Multiple Sclerosis; UIBC, Unsaturated Iron Binding Capacity; TIBC, Total Iron Binding Capacity.

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
