# Peer review of "The Role of Iron Metabolism in Fatigue, Depression, and Quality of Life in Multiple Sclerosis Patients"

_ijerph, 2020, doi:10.3390/ijerph17186818_

Round 1

Reviewer 1 Report

Thank you for the opportunity to review this manuscript. Overall, this is an important topic where the objective was to analyse the relationship between serum parameters of iron metabolism and the severity of fatigue, depressive symptoms and quality of life in MS patients.

I don't feel qualified to judge about the English language and style. There is good data in this study, however, the presentation and interpretation of the study need additional thought.   Here are some suggestions to improve the manuscript:

  1. Introduction

The introduction doesn´t provide sufficient background. You can include more relevant and new references (last years). Please, explain if “iron metabolism and the severity of fatigue, depressive symptoms and quality of life” have been studied in others articles and the review literature.

  1. Materials and Methods

Please, could you explain the design study?

How was the sample recruited? What were the inclusion and exclusion criteria for the subjects to participate in the study?

How was the sample size calculated, based on which other population or similar sample? I don’t know if it is representative.

On what dates was the study conducted? When were the measurements taken, at the beginning and at the end of what period?

What was the Cronbach's alpha in questionnaires used?

  1. Discussion

The discussion needs greater clarity on what the results showed. You should not repeat the numerical data of the results. Describe them in a descriptive way. For example: “The present study focused on the assessment of iron metabolism in 90 MS patients, including 213% with RRMS, 15% with SPMS, and 7% with PPMS”.
Start the discussion by reporting your own findings from the present study and then, after that, you put it in perspective of other available research and please, write the references. The discussion fails to clearly attempt to identify/explain reasons for results in this study that differ from other studies looking at similar outcomes.

Where are the Limitations of this study? Please include it.

Please include a Practical implications section before Conclusion section

References:

You need to properly review the bibliographic references at the end of the text. There is information missing in some references or it is not written in accordance with the journal's regulations.

Author Response

Dear Reviewer,

Thank you very much for your valuable comments, we tried to address them all by making corrections to our publication. We supplemented the introduction with current literature and systematized the discussion.

In response to the question about the size of the group, we decided on the number that is representative of the population of people with SI in Poland (about 45,000 patients), with an acceptable error of 10% due to the fact that patients with SI due to many contradictions and the lack of explanations about the pathogenesis and course of the disease, they are subjected to a lot of research, and therefore they are becoming less and less willing to participate in new research projects. In addition, we tried to meet the feature of the representative group of the study group by selecting respondents from 3 centers in Poland, one of which was sent to participants from all over the country. In addition, the heterogeneity of the population of MS patients remains the reason for the occurrence of variance in estimators, both in the patient population and in the study group.

We also supplemented the manuscript with information on limitations and the practical application of the results.

Attached I am sending the work with the changes introduced in two versions - tracking changes and with approved changes (to make it easier to read).

Best regards

Anna Knyszyńska

Reviewer 2 Report

General comments: 
In this paper, the authors investigated the relationship between some symptoms and iron metabolism in multiple sclerosis (MS) patients. They checked serum levels of iron, TIBC, UIBC, transferrin saturation, transferrin and ferritin in 90 MS patients. They also evaluated the patients’ fatigue, depression and quality of life according to the Fatigue Severity Scale, Beck Depression Inventory, and 25 Functional Assessment of Multiple Sclerosis. The results showed that ferritin levels were significantly correlated 26 with the severity of depressive symptoms and quality of life assessment in the MS patients. This is an interesting study. This study provided useful clinical findings about MS. My comments are as follows.

  1. The authors should investigate if iron supplementation can improve the level of ferrintin and recover some of the symptoms in MS patients.
  2. This study lakes comparison of iron metabolism between MS patients and healthy individuals.
  3. The authors may need to check the role of iron metabolism in serum antioxidant activity (for example, ORAC capacity of serum or urine 8-OHdG) in MS patients to explain the underline mechanism.
  4. There are some grammatical mistakes in the manuscript. The authors should ask a native English speaker or an English service to do proofreading.

Author Response

Dear Reviewer,

Thank you for your valuable comments, in our research we planned to analyze the parameters of oxidative stress in parallel, but the obtained funds prevented such an extension of the research. Patients included in the study declared no iron supplementation. In the future, however, we will certainly take all the comments into account when planning further research among patients with MS.

We made some changes to our manuscript, supplemented the reference literature with newer reports, and structured the discussion. I am sending the corrected work attached

Best regards

Anna Knyszyńska

Reviewer 3 Report

This is an interesting article about the potential relationship of iron blood parameters with fatigue and other mental factors of patients suffering from multiple sclerosis.

The experiment and statistical analysis are performed properly and the study group is large.

But the topic is difficult due to the fact the mental, subjective aspects of life are compared with the patients’ blood results. In addition, the group of patients is highly diversified according to their age. The results of correlation coefficient values are weak, and in fact, despite its statistical significance it is difficult to conclude about the relevance of obtained dependencies. Nevertheless, the authors' conclusions in detail justify these restrictions and present the current literature. 

Introduction:

There is a lack of explanation of the potential relationship between the physiological basis of fatigue and iron metabolism alone. Moreover, the introduction section should be supplemented with detailed information about the iron metabolism in MS syndrome e.g in the context of oxidative status. The conclusions contain this context, but some information should be present in the concept of this article.

Results 

Perhaps some graphical presentation would be more valuable and interesting than large tables, e.g in the form of a heatmap of a correlation coefficient, but this is only the Reviewer's opinion, not a necessity to change (tables contain the same information).

Author Response

Dear Reviewer,

Thank you very much for your valuable comments. We improved the introduction and systematized the discussion. Indeed, the results presented by us are placed in a large table, but this form seems to be the most accurate. Of course, perhaps the graphic form would be more readable, and we will keep that in mind when publishing further results.

Best regards

Anna Knyszyńska

Round 2

Reviewer 1 Report

Thank you very much for your response, the quality of your manuscript has been improved very much. Good Luck!!!